# MODEL COMPARISON FOR SEMANTIC GROUPING

## ABSTRACT

We introduce a probabilistic framework for quantifying the semantic similarity between two groups of embeddings. We formulate the task of semantic similarity as a model comparison task in which we contrast a generative model which jointly models two sentences versus one that does not. We illustrate how this framework can be used for the Semantic Textual Similarity tasks using clear assumptions about how the embeddings of words are generated. We apply information criteria based model comparison to overcome the shortcomings of Bayesian model comparison, whilst still penalising model complexity. We achieve competitive results by applying the proposed framework with an appropriate choice of likelihood on the STS datasets.

## 1 INTRODUCTION

The problem of Semantic Textual Similarity (STS), measuring how closely the meaning of one piece of text corresponds to that of another, has been studied in the hope of improving performance across various problems in Natural Language Processing (NLP), including information retrieval (Zheng & Callan, 2015). Recent progress in the generation of word embeddings (Mikolov et al., 2013) has allowed the encoding of words using distributed vector representations, which capture semantic information through their location in the learned embedding space. Despite the extent to which semantic relations between words are captured in this space, it remains a challenge to adapt these individual word embeddings to express semantic similarity between word groups, like documents, sentences, and other textual formats.

Recent methods for STS rely on additive composition of word vectors (Arora et al., 2016; Blacoe & Lapata, 2012; Mitchell & Lapata, 2008; 2010) or deep learning architectures (Kiros et al., 2015), which summarise a sentence through a single embedding. The resulting sentence vectors are then compared using cosine similarity — a choice that is not theoretically justified, stemming only from the fact that cosine similarity gives good empirical results. It is difficult for a practitioner to utilize word vectors efficiently if the underlying assumptions in the similarity measure are not well understood.

The main contribution of our work is the proposal of a framework that addresses these issues by explicitly deriving the similarity measure through a chosen generative model of embeddings, instead of empirically motivating it. Via this design process, a practitioner can encode suitable assumptions and constraints that may be favourable to the application of interest. Furthermore, this framework puts forward a new research direction that could help understand semantic similarity, in which practitioners can study suitable embedding distributions and assess how these perform.

The secondary contribution of our work is a derivation of a similarity measure that performs well in an online setting[1]. Online settings are both practical and key to use-cases that involve information retrieval in dialogue systems. For example, in a chat-bot application new queries will arrive one at a time and methods such as the one proposed in Arora et al. (2016) will not perform as strongly as they do on the benchmark datasets. This is because one cannot perform the required data pre-processing on the entire query dataset, which will not be available a priori in online settings. Whilst our framework produces an online similarity metric, it remains competitive to offline methods such as Arora et al. (2016). We achieve results comparable to Arora et al. (2016) on the STS dataset in

---

[1] The difference between an online and an offline setting is whether or not one has access to the entire dataset at evaluation time. That is, in an online setting, one has access to only a single query pair at a time.

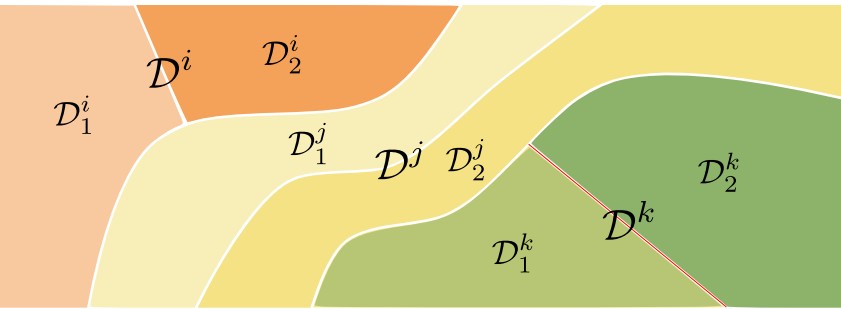

Figure 1: The generation of embeddings of words in a sentence through a latent variable (the semantic group). Above we visualize the semantic group space of embeddings (with $\mathcal{D}^i, \mathcal{D}^j, \mathcal{D}^k$ representing different semantic groups).

$\mathcal{O}(nd)$ time — compared to the $\mathcal{O}(nd^2)$ average complexity of their method (where $n$ is the number of words in a sentence, and $d$ is the embedding size).

## 2 BACKGROUND

The compositional nature of distributed representations demonstrated in Mikolov et al. (2013) and Pennington et al. (2014) indicate the presence of semantic groups in the representation space of word embeddings; an idea which has been further explored in Athiwaratkun & Wilson (2017). Under this assumption, the task of semantic similarity can be formulated as the following question: "Are the two sentences (as groups of words) partitions of the same semantic group?" Figure 1 illustrates this concept. Utilising this rephrasing, this work formulates the task of semantic similarity between two arbitrary groups of objects as a model comparison problem. Taking inspiration from Ghahramani & Heller (2006) and Marshall et al. (2006) we propose the generative models for groups (e.g. sentences) $\mathcal{D}_1, \mathcal{D}_2$ seen in Figure 2.

The Bayes Factor for this graphical model is then formally defined as

$$\mathbf{sim}(\mathcal{D}_1, \mathcal{D}_2) = \log \frac{p(\mathcal{D}_1, \mathcal{D}_2|\mathcal{M}_1)}{p(\mathcal{D}_1, \mathcal{D}_2|\mathcal{M}_2)} = \log \frac{p(\mathcal{D}_1, \mathcal{D}_2|\mathcal{M}_1)}{p(\mathcal{D}_1|\mathcal{M}_2)p(\mathcal{D}_2|\mathcal{M}_2)}. \tag{1}$$

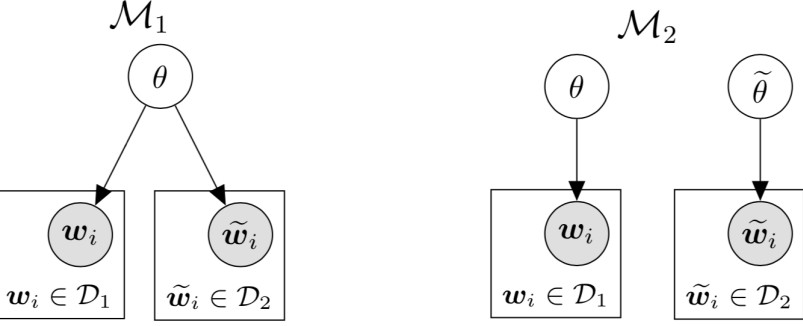

Figure 2: On the left, $\mathcal{M}_1$ assumes that both datasets are generated i.i.d. from the same parametric distribution. On the right, $\mathcal{M}_2$ assumes that the datasets are generated i.i.d. from distinct parametric distributions.

To obtain the evidence $p(\mathcal{D}|\mathcal{M}_i)$ the parameters of the model must be marginalised

$$p(\mathcal{D}_1, \mathcal{D}_2|\mathcal{M}_1) = \int p(\mathcal{D}_1, \mathcal{D}_2|\theta)p(\theta)d\theta = \int \prod_{\boldsymbol{w}_k \in \mathcal{D}_1 \oplus \mathcal{D}_2} p(\boldsymbol{w}_k|\theta)p(\theta)d\theta,$$

$$p(\mathcal{D}_i|\mathcal{M}_2) = \int \prod_{\boldsymbol{w}_k \in \mathcal{D}_i} p(\boldsymbol{w}_k|\theta)p(\theta)d\theta,$$

where $\oplus$ denotes concatenation.

Computing the semantic similarity score of the two groups $\mathcal{D}_1, \mathcal{D}_2$ under the Bayesian framework requires selecting a reasonable model likelihood $p(\boldsymbol{w}_k|\theta)$, prior density on the parameters $p(\theta)$, and computing the marginal evidence specified above. Computing the evidence can be computationally intensive and usually requires approximation. What is more, the Bayes factor is very sensitive to the choice of prior and can result in estimates that heavily underfit the data (especially under a vague prior, as shown in Appendix E), having the tendency to select the simpler model; which is argued further by M. S. Bartlett (1957) and Akaike et al. (1981). This is handled in Ghahramani & Heller (2006) by using the empirical Bayes procedure, a shortcoming of which is the issue of double counting (Berger, 2000) and thus being prone to over-fitting. We address these issues by choosing to work with information criteria based model comparison as opposed to using the Bayes factor. The details are described in Section 3.

We are not aware of any prior work on sentence similarity that uses our approach. We employ a generative model for sentences similar to Arora et al. (2016), but our contribution differs from theirs in that our similarity is based on the aforementioned model comparison test, whilst theirs is based on the inner product of sentence embeddings derived as maximum likelihood estimators. The method by Marshall et al. (2006) applies the same score as in Equation 1 in the context of merging datasets whilst we focus on information retrieval and semantic textual similarity.

## 3 METHODOLOGY

We address the shortcomings of the Bayes Factor described in Section 2 by proposing model comparison criteria that minimise Kullback-Leibler (KL) divergence (denoted in equations as $D_{KL}$) across a candidate set of models. This results in a penalised likelihood ratio test which gives competitive results. This approach relies on Information Theoretic Criteria to design a log-ratio-like test that is prior free and robust to overfitting.

We seek to compare models $\mathcal{M}_1$ and $\mathcal{M}_2$ using Information Criteria (IC) to assess the goodness of fit of each model. There are multiple IC used for model selection, each with different settings to which they are better suited. The IC which we will be working with have the general form

$$\text{IC}(\mathcal{D}, \mathcal{M}) = -\left(\alpha \mathcal{L}(\hat{\boldsymbol{\theta}}|\mathcal{D}, \mathcal{M}) + \Omega(\mathcal{D}, \mathcal{M})\right),$$

where $\mathcal{L}(\hat{\boldsymbol{\theta}}|\mathcal{D}, \mathcal{M}) = \sum_i^n \mathcal{L}(\hat{\boldsymbol{\theta}}|\boldsymbol{w}_i, \mathcal{M})$ is the maximised value of the log likelihood function for model $\mathcal{M}$, $\alpha$ is a scalar derived for each IC, and $\Omega(\mathcal{D}, \mathcal{M})$ represents a model complexity penalty term which is model and IC specific. Using this general formulation for the involved information criteria yields the following similarity score:

$$\textbf{sim}(\mathcal{D}_1, \mathcal{D}_2) = -\text{IC}(\{\mathcal{D}_1, \mathcal{D}_2\}, \mathcal{M}_1) + \text{IC}(\{\mathcal{D}_1, \mathcal{D}_2\}, \mathcal{M}_2)$$

$$= \alpha\left(\hat{\mathcal{L}}(\hat{\boldsymbol{\theta}}_{1,2}|\mathcal{M}_1) - (\hat{\mathcal{L}}(\hat{\boldsymbol{\theta}}_1|\mathcal{M}_2) + \hat{\mathcal{L}}(\hat{\boldsymbol{\theta}}_2|\mathcal{M}_2))\right)$$

$$- \Omega(\{\mathcal{D}_1, \mathcal{D}_2\}, \mathcal{M}_1) + \Omega(\{\mathcal{D}_1, \mathcal{D}_2\}, \mathcal{M}_2).$$

The paragraph surrounding Equation (9) of Tenenbaum & Griffiths (2001) (Tversky's contrast model) describes what a good similarity is from a cognitive science perspective. Interestingly, the semantic interpretation of the similarity we have derived is similar to the one described in that work. We want to contrast the commonalities of the two datasets (through shared parameters) to the distinctive features of each dataset (through independently fit parameters).

Examples of these criteria can be put into two broad classes. The Bayesian Information Criterion (BIC) is an example of an IC that approximates the model evidence directly, as defined in

Schwarz et al. (1978). Empirically it has been shown that the BIC is likely to underfit the data, especially when the number of samples is small (Dziak et al., 2012). We provide additional empirical results in Appendix E, showing BIC is not a good fit for the STS task as sentences contain a relatively small number of words (samples). Thus we focus on the second class - Information Theoretic Criteria.

## 3.1 INFORMATION THEORETIC CRITERIA

The Information Theoretic Criteria (ITC) are a family of model selection criteria. The task they address is evaluating the expected quality of an estimated model specified by $\mathcal{L}(\hat{\boldsymbol{\theta}}|\boldsymbol{w})$ when it is used to generate unseen data from the true distribution $G(\boldsymbol{w})$, as defined in Konishi & Kitagawa (2008b). This family of criteria perform this evaluation by using the KL divergence between the true model $G(\boldsymbol{w})$ and the fitted model $\mathcal{L}(\hat{\boldsymbol{\theta}}|\boldsymbol{w})$, with the aim of selecting the model (from a given set of models) that minimizes the quantity

$$D_{KL}\left(G(\boldsymbol{w})\big|\big|p(\boldsymbol{w}|\hat{\boldsymbol{\theta}})\right) = \mathbb{E}_G\left[\ln\frac{G(\boldsymbol{w})}{p(\boldsymbol{w}|\hat{\boldsymbol{\theta}})}\right] = -H_G(\boldsymbol{w}) - \mathbb{E}_G\left[\ln p(\boldsymbol{w}|\hat{\boldsymbol{\theta}})\right].$$

The entropy of the true model $H_G(\boldsymbol{w})$ will remain constant across different likelihoods. Thus, the quantity of interest in the definition of the information criterion under consideration is given by the expected log likelihood under the true model $\mathbb{E}_G[\ln p(\boldsymbol{w}|\hat{\boldsymbol{\theta}})]$. The goal then, is to find a good estimator for this quantity. One such estimator is given by the normalized maximum likelihood:

$$\mathbb{E}_{\hat{G}}\left[\ln p(\boldsymbol{w}|\hat{\boldsymbol{\theta}})\right] = \frac{1}{n}\sum_{i=1}^n \ln p(\boldsymbol{w}_i|\hat{\boldsymbol{\theta}}),$$

where $\hat{G}$ represents the empirical distribution. This estimator introduces a bias that varies with respect to the dimension of the model's parameter vector $\boldsymbol{\theta}$ and requires a correction in order to carry out a fair comparison of information criteria between models. A model specific correction is derived resulting in the following IC, called Takeuchi Information Criterion (TIC) in Takeuchi (1976)

$$\hat{\boldsymbol{J}} = -\frac{1}{n}\sum_{i=1}^n \nabla_{\boldsymbol{\theta}}^2 \mathcal{L}(\boldsymbol{\theta}|\boldsymbol{w}_i)\bigg|_{\boldsymbol{\theta}=\hat{\boldsymbol{\theta}}}, \qquad \hat{\boldsymbol{\mathcal{I}}}, = \frac{1}{n}\sum_{i=1}^n \nabla_{\boldsymbol{\theta}}\mathcal{L}(\boldsymbol{\theta}|\boldsymbol{w}_i)\nabla_{\boldsymbol{\theta}}\mathcal{L}^\top(\boldsymbol{\theta}|\boldsymbol{w}_i)\bigg|_{\boldsymbol{\theta}=\hat{\boldsymbol{\theta}}},$$

$$\text{TIC}(\mathcal{D}, \mathcal{M}) = -2\left(\mathcal{L}(\hat{\boldsymbol{\theta}}|\mathcal{D}, \mathcal{M}) - \text{tr}\left(\hat{\boldsymbol{\mathcal{I}}}\hat{\boldsymbol{J}}^{-1}\right)\right).$$

Then, under the TIC we have the following similarity (full derivation in Appendix A)

$$\mathbf{sim}(\mathcal{D}_1, \mathcal{D}_2) = 2\bigg(\mathcal{L}(\hat{\boldsymbol{\theta}}_{1,2}|\mathcal{M}_1) - \mathcal{L}(\hat{\boldsymbol{\theta}}_1|\mathcal{M}_2) - \mathcal{L}(\hat{\boldsymbol{\theta}}_2|\mathcal{M}_2)$$

$$- \text{tr}\left(\hat{\boldsymbol{\mathcal{I}}}_{1,2}\hat{\boldsymbol{J}}_{1,2}^{-1}\right) + \text{tr}\left(\hat{\boldsymbol{\mathcal{I}}}_1\hat{\boldsymbol{J}}_1^{-1}\right) + \text{tr}\left(\hat{\boldsymbol{\mathcal{I}}}_2\hat{\boldsymbol{J}}_2^{-1}\right)\bigg).$$

For the case where we assume our model has the same parametric form as the true model and as $n \to \infty$, the equality $\hat{\boldsymbol{\mathcal{I}}} = \hat{\boldsymbol{J}}$ holds resulting in a penalty of $\text{tr}\left(\hat{\boldsymbol{\mathcal{I}}}\hat{\boldsymbol{J}}^{-1}\right) = \text{tr}(\boldsymbol{I}_k) = k$ where $k$ is the number of model parameters, as shown in Konishi & Kitagawa (2008a). This results in the Akaike Information Criterion (AIC) (Akaike, 1974)

$$\text{AIC}(\mathcal{D}, \mathcal{M}) = -2(\mathcal{L}(\hat{\boldsymbol{\theta}}|\mathcal{D}, \mathcal{M}) - k).$$

The AIC simplification of TIC relies on several assumptions that only hold true in the big data limit. AIC also assumes our model $\mathcal{M}$ has the same parametric form as the true model (Konishi & Kitagawa (2008a)). In general, TIC is a more robust approximation. However, as shown in Appendix F, for models with a high number of parameters, TIC may prove unstable and thus AIC will generally perform better. In this study we will consider and contrast both.

## 4 LIKELIHOODS

In this section we will illustrate how to derive a similarity score under our ITC framework by choosing a likelihood function that incorporates our prior assumptions about the generating process of the data. Adopting the viewpoint of a practitioner, we would like to compare the performance of two models — one that ignores word embedding magnitude, and one that makes use of it. Our modelling choices for each assumption are the von Mises-Fisher (vMF) and Gaussian likelihoods respectively. The comparison between the two likelihoods we provide in Section 5 provides empirical evidence as to which approach is better suited to modelling word embeddings. The rest of this section outlines the details of each of the two modelling choices.

### 4.1 VON MISES-FISHER LIKELIHOOD

Word embeddings are $d$ dimensional vectors of real numbers, that are traditionally learned by optimizing a dot product between target words and context vectors (Mikolov et al., 2013). Due to this training setup, cosine similarity is often used to measure the semantic similarity of words in various information retrieval tasks. Thus, we want to explore a distribution induced by the cosine similarity measure. We model our embeddings as vectors lying on the surface of the $d - 1$ dimensional unit hyper-sphere $\boldsymbol{w} \in \mathbb{S}^{d-1}$ and i.i.d. according to a vMF likelihood (Fisher et al. (1993))

$$p(\boldsymbol{w}|\boldsymbol{\mu}, \kappa) = \frac{\kappa^{\frac{d}{2}-1}}{(2\pi)^{\frac{d}{2}} I_{\frac{d}{2}-1}(\kappa)} \exp\left(\kappa\boldsymbol{\mu}^\top \boldsymbol{w}\right) = \frac{1}{Z(\kappa)} \exp\left(\kappa\boldsymbol{\mu}^\top \boldsymbol{w}\right),$$

where $\boldsymbol{\mu}$ is the mean direction vector and $\kappa$ is the concentration, with supports $||\boldsymbol{\mu}|| = ||\boldsymbol{w}|| = 1$, $\kappa \geq 0$. The term $I_\nu(\kappa)$ Corresponds to a modified Bessel function of the first kind with order $\nu$.

In this work we reparameterise the random variable to polar hypersphericals $\boldsymbol{w}(\boldsymbol{\phi})$ ($\boldsymbol{\phi} = (\phi_1, ..., \phi_{d-1})^\top$) as adopted in Mabdia (1975). Further details can be found in Appendix B.

The first and second order partial derivatives of the vMF log likelihood are derived in Appendix C.

We prove (in Appendix C) that the mixed derivatives of the vMF log likelihood are a constant (with respect to $\boldsymbol{\phi}$) times $\partial\mathcal{L}(\boldsymbol{\theta}, \kappa|\boldsymbol{\phi})/\partial\theta_k$. Thus, evaluated at the MLE, these entries are zero. Assuming $l < k$

$$\left.\frac{\partial^2\mathcal{L}(\boldsymbol{\theta}, \kappa|\mathcal{D})}{\partial\kappa\partial\theta_k}\right|_{\boldsymbol{\theta}=\hat{\boldsymbol{\theta}}, \kappa=\hat{\kappa}} = \left.\frac{\partial^2\mathcal{L}(\boldsymbol{\theta}, \kappa|\mathcal{D})}{\partial\theta_k\partial\theta_l}\right|_{\boldsymbol{\theta}=\hat{\boldsymbol{\theta}}, \kappa=\hat{\kappa}} = 0.$$

Thus, $\hat{\boldsymbol{J}}$ is a diagonal matrix, with diagonal $\boldsymbol{j} = (\hat{J}_{11}, ..., \hat{J}_{dd})^\top$

$$\text{tr}(\hat{\boldsymbol{\mathcal{I}}}\hat{\boldsymbol{J}}^{-1}) = \sum_{i=1}^d \hat{J}_{ii}^{-1}\hat{\mathcal{I}}_{ii} = \hat{J}_{11}^{-1}\left(\frac{\partial}{\partial\kappa}\mathcal{L}(\boldsymbol{\theta}, \kappa|\mathcal{D})\right)^2 + \sum_{i=2}^d \hat{J}_{ii}^{-1}\left(\frac{\partial}{\partial\theta_{i-1}}\mathcal{L}(\boldsymbol{\theta}, \kappa|\mathcal{D})\right)^2. \quad (2)$$

This quantity only requires $\mathcal{O}(nd)$ operations to compute and thus does not increase the asymptotic complexity of the algorithm.

The closed form of the similarity measure for two sentences $\mathcal{D}_1, \mathcal{D}_2$ of length $m$ and $l$ respectively under this model is then

$$\begin{aligned}\text{sim}(\mathcal{D}_1, \mathcal{D}_2) =& (m+l)\hat{\kappa}_{1,2}\bar{R}_{1,2} - m\hat{\kappa}_1\bar{R}_1 - l\hat{\kappa}_2\bar{R}_2 \\ & - (m+l)\log Z(\hat{\kappa}_{1,2}) + m\log Z(\hat{\kappa}_1) + l\log Z(\hat{\kappa}_2) \\ & - \text{tr}(\hat{\boldsymbol{\mathcal{I}}}_{1,2}\hat{\boldsymbol{J}}_{1,2}^{-1}) + \text{tr}(\hat{\boldsymbol{\mathcal{I}}}_1\hat{\boldsymbol{J}}_1^{-1}) + \text{tr}(\hat{\boldsymbol{\mathcal{I}}}_2\hat{\boldsymbol{J}}_2^{-1}),\end{aligned}$$

where the Jacobian terms (from the reparametrisation) cancel out. The subscripts indicate the sentence, with $1, 2$ meaning the concatenation of the two sentences.

### 4.2 GAUSSIAN LIKELIHOOD

Schakel & Wilson (2015) show that some frequency information is contained in the magnitude of word embeddings. This motivates a choice of a likelihood function that is not constrained to the unit

hyper-sphere and possibly the simplest such choice is the Gaussian likelihood. Due to the small size of sentences[2] we choose a diagonal covariance Gaussian models. The Gaussian likelihood is then

$$p\left(\boldsymbol{w}|\boldsymbol{\mu}, \boldsymbol{\Sigma}\right) = \mathcal{N}\left(\boldsymbol{w}|\boldsymbol{\mu}, \boldsymbol{\Sigma}\right),$$

where $\boldsymbol{\Sigma}$ is a diagonal matrix.

As with the vMF likelihood, we prove in Appendix D that the Hessian of the log likelihood evaluated at the MLE is diagonal. The TIC correction is a sum of $O(d)$ terms, similar to the form in Equation 2. This can be nicely written in terms of biased sample kurtosis (denoted $\hat{\boldsymbol{\kappa}}$)

$$\mathrm{tr}(\hat{\boldsymbol{\mathcal{I}}}\hat{\boldsymbol{J}}^{-1}) = \frac{1}{2}\left(d + \sum_{i=1}^{d}\frac{(\hat{\boldsymbol{\mu}}_4)_i}{\hat{\sigma}_i^4}\right) = \frac{d}{2} + \sum_{i=1}^{d}\frac{\hat{\kappa}_i}{2}$$

The closed form of the similarity measure for two sentences $\mathcal{D}_1, \mathcal{D}_2$ of length $m$ and $l$ respectively under this model is then

$$\mathbf{sim}(\mathcal{D}_1, \mathcal{D}_2) = \sum_{i=1}^{d} -(m+l)\ln(\hat{\boldsymbol{\sigma}}_{1,2})_i + m\ln(\hat{\boldsymbol{\sigma}}_1)_i + l\ln(\hat{\boldsymbol{\sigma}}_2)_i$$

$$+ \frac{d}{2} + \frac{1}{2}\sum_{i=1}^{d} -(\hat{\boldsymbol{\kappa}}_{1,2})_i + (\hat{\boldsymbol{\kappa}}_1)_i + (\hat{\boldsymbol{\kappa}}_2)_i$$

where the subscripts indicate the sentence, with $1, 2$ meaning the concatenation of the two sentences.

## 5 EXPERIMENTS

We assess our methods' performance on the Semantic Textual Similarity (STS) datasets[3] (Agirre et al., 2012; 2013; 2014; 2015; 2016). The objective of these tasks is to estimate the similarity between two given sentences, validated against human scores. In our experiments, we assess on the pre-trained GloVe (Pennington et al., 2014), FastText (Bojanowski et al., 2016), and Word2Vec (Mikolov et al., 2013) word embeddings. For the vMF distribution, we centre the distribution of word embeddings and normalize them to be of length 1. Some of the sentences are left with a single word after querying the word embeddings, making the MLE of the $\kappa$ parameter of the vMF and $\boldsymbol{\Sigma}$ parameter of the Gaussian ill-defined. We overcome this issue by padding each sentence with an arbitrary embedding of a word or punctuation symbol (i.e. '.' or '?').

We first compare our methods: vMF likelihood with TIC correction (vMF+TIC) and diagonal Gaussian likelihood with AIC correction (Diag+AIC) against each other. Then, the better method is compared against mean word vector (MWV), word mover's distance (WMD) (Kusner et al., 2015) [4], smooth inverse frequency (SIF), and SIF with principal component removal as defined in Arora et al. (2016) [5]. We re-ran these models under our experimental setup, to ensure a fair comparison. The metric used is the average Spearman correlation score over each dataset, weighted by the number of sentences. The choice of Spearman correlation is given by its non-parametric nature (assumes no distribution over the scores), as well as measuring any monotonic relationship between the two compared quantities.

### 5.1 EMBEDDING MAGNITUDE

A practitioner may want to learn more about a given set of word embeddings, and the way these embeddings were trained may not allow the user to understand the importance of certain features — say embedding magnitude. This is where our framework can be used to help build intuition by comparing a likelihood that implicitly incorporates embedding magnitude to one that does not.

---

[2]The covariance matrix of $n$ samples with $d$ dimensions such that $n < d$ results in a low rank matrix.

[3]The STS13 dataset does not include the proprietary SMT dataset that was available with the original release of STS.

[4]https://github.com/mkusner/wmd

[5]https://github.com/PrincetonML/SIF

| Embedding | Method | STS12 | STS13 | STS14 | STS15 | STS16 | Average |
|-----------|--------|-------|-------|-------|-------|-------|---------|
| FastText | vMF+TIC | 0.5578 | 0.5532 | 0.5897 | 0.6660 | 0.6596 | 0.6023 |
|          | Diag+AIC | **0.6193** | **0.6335** | **0.6721** | **0.7328** | **0.7518** | **0.6765** |
| GloVe | vMF+TIC | 0.5516 | 0.5873 | 0.5865 | 0.6605 | 0.6083 | 0.5977 |
|       | Diag+AIC | **0.6031** | **0.6132** | **0.6445** | **0.7171** | **0.7346** | **0.6564** |
| Word2Vec | vMF+TIC | 0.5161 | 0.4997 | 0.5661 | 0.6477 | 0.5984 | 0.5683 |
|          | Diag+AIC | **0.5630** | **0.5799** | **0.6291** | **0.6951** | **0.6701** | **0.6265** |

Table 1: Comparison of Spearman correlations on the STS datasets between the two similarity measures we introduce in the text. The average is weighted according to dataset size.

We present the comparison between the similarities derived from the vMF and Gaussian likelihoods in Table 5.1. We note that the Gaussian is a much better modelling choice, beating the vMF on every dataset by a margin of at least 0.05 (5%) on average, with each of the three word embeddings. This is strong evidence that the information encoded in an embedding's magnitude is useful for tasks such as semantic similarity. This further motivates the conjecture that frequency information is contained in word embedding magnitude, as explored in Schakel & Wilson (2015).

## 5.2 ONLINE SCENARIO

In Section 1, it was mentioned that in an online setting, one cannot perform the principal component removal described in Arora et al. (2016), as that pre-processing requires access to the entire query dataset a priori. This scenario arises in the context of information retrieval, e.g. when creating a chat-bot.

The comparison against the baseline methods are presented in Table 5.2. We are able to out-perform the standard weighting induced by MWV, as well as the WMD approach on all datasets, with each of the three word embeddings considered. We outperform SIF using the GloVe embeddings by 0.0278 (2.78%), effectively tie when using the FastText embeddings with a difference of 0.0021 (0.21%) and we are marginally below using Word2Vec by 0.0028 (0.28%). The latter two differences are small enough to be considered insignificant.

## 5.3 OFFLINE SCENARIO

There are use-cases in which the entire dataset of sentences is available at evaluation time — for example, in clustering applications. For this scenario, we compare against the SIF weightings,

| Embedding | Method | STS12 | STS13 | STS14 | STS15 | STS16 | Average |
|-----------|--------|-------|-------|-------|-------|-------|---------|
| FastText | Diag+AIC | **0.6193** | 0.6335 | **0.6721** | 0.7328 | **0.7518** | **0.6765** |
|          | SIF | 0.6003 | **0.6921** | **0.6729** | **0.7473** | 0.7012 | **0.6777** |
|          | MWV | 0.5994 | 0.6494 | 0.6473 | 0.7114 | 0.6814 | 0.6542 |
|          | WMD | 0.5576 | 0.5146 | 0.5915 | 0.6800 | 0.6402 | 0.5997 |
| GloVe | Diag+AIC | **0.6031** | 0.6132 | **0.6445** | **0.7171** | **0.7346** | **0.6564** |
|       | SIF | 0.5754 | **0.6269** | 0.6113 | 0.6899 | 0.6699 | 0.6286 |
|       | MWV | 0.5526 | 0.5643 | 0.5625 | 0.6314 | 0.5804 | 0.5784 |
|       | WMD | 0.5516 | 0.5007 | 0.5811 | 0.6704 | 0.6246 | 0.5896 |
| Word2Vec | Diag+AIC | **0.5630** | 0.5799 | 0.6291 | 0.6951 | **0.6701** | **0.6265** |
|          | SIF | 0.5400 | **0.6352** | **0.6386** | **0.7027** | 0.6413 | **0.6293** |
|          | MWV | 0.5329 | 0.5796 | 0.5886 | 0.6385 | 0.5776 | 0.5846 |
|          | WMD | 0.5162 | 0.4966 | 0.5778 | 0.6630 | 0.6022 | 0.5755 |

Table 2: Comparison of Spearman correlations on the STS datasets between our best model (Diag+AIC) and SIF,WMD and MWV for three different word vectors. The average is weighted according to dataset size.

| Word vectors | Method | STS12 | STS13 | STS14 | STS15 | STS16 | Average |
|---|---|---|---|---|---|---|---|
| FastText | Diag+AIC | **0.6193** | 0.6335 | 0.6721 | 0.7328 | **0.7518** | 0.6765 |
| | SIF+PCA | 0.5893 | **0.7121** | **0.6790** | **0.7498** | 0.7142 | **0.6810** |
| GloVe | Diag+AIC | **0.6031** | 0.6132 | 0.6445 | **0.7171** | **0.7346** | **0.6564** |
| | SIF+PCA | 0.5681 | **0.6844** | **0.6546** | 0.7166 | 0.6931 | 0.6552 |
| Word2Vec | Diag+AIC | **0.5630** | 0.5799 | 0.6291 | 0.6951 | **0.6701** | 0.6265 |
| | SIF+PCA | 0.5324 | **0.6486** | **0.6510** | **0.7031** | 0.6609 | **0.6347** |

Table 3: Comparison of Spearman correlations on the STS datasets between our best model (Diag+AIC) and SIF+PCA for three different word vectors.

augmented with the additional pre-processing technique seen in Arora et al. (2016). We need only consider this baseline, as it outperforms all others by a large margin.

The results are shown in Table 5.3. We remaing competitive with SIF+PCA on all three word embeddings, being able to match very closely on GloVe embeddings. On the FastText and Word2Vec embeddings, our method is less than 0.01 (1%) lower on average than SIF+PCA.

## 6 CONCLUSION

We've presented a new approach to similarity measurement that achieves competitive performance to standard methods in both online and offline settings. Our method requires a set of clear choices — model, likelihood and information criterion. From that, a comparison framework is naturally derived, which supplies us with a statistically justified similarity measure (by utilizing ITC to reduce the resulting model-comparison bias). This framework is suitable for a variety of modelling scenarios, due to the freedom in specifying the generative process. The graphical model we employ is adaptable to encode structural dependencies beyond the i.i.d. data-generating process we have assumed throughout this study — for example, an auto-regressive (sequential) model may be assumed if the practitioner suspects that word order matters (i.e. compare "Does she want to get pregnant?" to "She does want to get pregnant.")

In this study, we conjecture that the von Mises-Fisher distribution lends itself to representing word embeddings well, if their magnitude is disregarded and a unimodal distribution over individual sentences is assumed. Relaxing the former assumption, we also model word embeddings with a Gaussian likelihood. As this improves results, this is empirical evidence that word embedding magnitude carries relevant information, which agrees with prior intuition built from Schakel & Wilson (2015). We hope that this framework could be a stepping stone in using more complex and accurate generative models of text to assess semantic similarity. For example, relaxing the assumption of unimodality is an interesting area for future research.

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

## A  DERIVATION OF TIC FOR $\mathcal{M}_2$

The MLE for $p(\phi_{1,2}|\boldsymbol{\theta}, \mathcal{M}_2)$ can be derived by simply estimating the separate MLE solutions for $p(\phi_1|\boldsymbol{\theta}_1, \mathcal{M}_2)$ and $p(\phi_2|\boldsymbol{\theta}_2, \mathcal{M}_2)$. What is not as obvious is that the penalty term follows the estimation pattern.

Gradient vectors for $\mathcal{M}_2$ are given by (where $\oplus$ is concatenation)

$$\nabla_{\boldsymbol{\theta}}\mathcal{L}(\boldsymbol{\theta}|\phi_{1,2}) = \nabla_{\boldsymbol{\theta}_1}\mathcal{L}(\boldsymbol{\theta}_1|\phi_1) \oplus \nabla_{\boldsymbol{\theta}_2}\mathcal{L}(\boldsymbol{\theta}_2|\phi_2),$$

and Hessian results in a block diagonal matrix

$$\nabla_{\boldsymbol{\theta}}^2\mathcal{L}(\boldsymbol{\theta}|\phi_{1,2}) = \begin{bmatrix} \nabla_{\boldsymbol{\theta}_1}^2\mathcal{L}(\boldsymbol{\theta}_1|\phi_1) & \mathbf{0} \\ \mathbf{0} & \nabla_{\boldsymbol{\theta}_2}^2\mathcal{L}(\boldsymbol{\theta}_2|\phi_2) \end{bmatrix},$$

with inverse

$$\nabla_{\boldsymbol{\theta}}^2\mathcal{L}(\boldsymbol{\theta}|\phi_{1,2})^{-1} = \begin{bmatrix} \nabla_{\boldsymbol{\theta}_1}^2\mathcal{L}(\boldsymbol{\theta}_1|\phi_1)^{-1} & \mathbf{0} \\ \mathbf{0} & \nabla_{\boldsymbol{\theta}_2}^2\mathcal{L}(\boldsymbol{\theta}_2|\phi_2)^{-1} \end{bmatrix}.$$

Computing $\mathrm{tr}(\hat{\boldsymbol{\mathcal{I}}}\hat{\boldsymbol{J}}^{-1})$ then yields

$$\begin{aligned}
\mathrm{tr}(\hat{\boldsymbol{\mathcal{I}}}\hat{\boldsymbol{J}}^{-1}) &= \mathrm{tr}\left(\nabla_{\boldsymbol{\theta}}\mathcal{L}(\boldsymbol{\theta}|\phi_{1,2})\nabla_{\boldsymbol{\theta}}\mathcal{L}(\boldsymbol{\theta}|\phi_{1,2})^{\top}\nabla_{\boldsymbol{\theta}}^2\mathcal{L}(\boldsymbol{\theta}|\phi_{1,2})^{-1}\right) \\
&= \mathrm{tr}\left(\begin{bmatrix} \hat{\boldsymbol{\mathcal{I}}}_{11}\nabla_{\boldsymbol{\theta}_1}^2\mathcal{L}(\boldsymbol{\theta}_1|\phi_1)^{-1} & \hat{\boldsymbol{\mathcal{I}}}_{12}\nabla_{\boldsymbol{\theta}_2}^2\mathcal{L}(\boldsymbol{\theta}_2|\phi_2)^{-1} \\ \hat{\boldsymbol{\mathcal{I}}}_{12}\nabla_{\boldsymbol{\theta}_1}^2\mathcal{L}(\boldsymbol{\theta}_1|\phi_1)^{-1} & \hat{\boldsymbol{\mathcal{I}}}_{22}\nabla_{\boldsymbol{\theta}_2}^2\mathcal{L}(\boldsymbol{\theta}_2|\phi_2)^{-1} \end{bmatrix}\right) \\
&= \mathrm{tr}\left(\hat{\boldsymbol{\mathcal{I}}}_{11}\nabla_{\boldsymbol{\theta}_1}^2\mathcal{L}(\boldsymbol{\theta}_1|\phi_1)^{-1}\right) + \mathrm{tr}\left(\hat{\boldsymbol{\mathcal{I}}}_{22}\nabla_{\boldsymbol{\theta}_2}^2\mathcal{L}(\boldsymbol{\theta}_2|\phi_2)^{-1}\right) \\
&= \mathrm{tr}\left(\hat{\boldsymbol{\mathcal{I}}}_1\hat{\boldsymbol{J}}_1^{-1}\right) + \mathrm{tr}\left(\hat{\boldsymbol{\mathcal{I}}}_2\hat{\boldsymbol{J}}_2^{-1}\right).
\end{aligned}$$

## B  REPARAMETRISATION OF THE vMF DISTRIBUTION

We reparametrise the random variable to polar hypersphericals $\boldsymbol{w}(\boldsymbol{\phi})$ ($\boldsymbol{\phi} = (\phi_1, ..., \phi_{d-1})^{\top}$) as adopted in Mabdia (1975)

$$p(\boldsymbol{\phi}|\boldsymbol{\theta}, \kappa) = \left(\frac{\kappa^{\frac{d}{2}-1}}{(2\pi)^{\frac{d}{2}}I_{\frac{d}{2}-1}(\kappa)}\right)\left|\frac{\partial\boldsymbol{w}}{\partial\boldsymbol{\phi}}\right|\exp\left(\kappa\boldsymbol{\mu}(\boldsymbol{\theta})^{\top}\boldsymbol{w}(\boldsymbol{\phi})\right),$$

where

$$w_i(\boldsymbol{\phi}) = ((1-\delta_{id})\cos\phi_i + \delta_{id})\prod_{k=1}^{i-1}\sin\phi_k, \quad \mu_i(\boldsymbol{\theta}) = ((1-\delta_{id})\cos\theta_i + \delta_{id})\prod_{k=1}^{i-1}\sin\theta_k,$$

$$\left|\frac{\partial\boldsymbol{w}}{\partial\boldsymbol{\phi}}\right| = \prod_{k=1}^{d-2}(\sin\phi_k)^{d-k-1}.$$

This reparametrisation simplifies the calculation of partial derivatives. The maxima of the likelihood remains unchanged since $|\partial\boldsymbol{w}/\partial\boldsymbol{\phi}|$ does not depend on $\boldsymbol{\theta}$ thus the MLE estimate in the hypershperical coordinates parametrisation is given by applying the map from the cartesian MLE to the polars.

$$\hat{\boldsymbol{\mu}} = \frac{\sum_{i=1}^n\boldsymbol{w}_i}{||\sum_{i=1}^n\boldsymbol{w}_i||}, \quad \hat{\boldsymbol{\theta}} = \boldsymbol{\mu}^{-1}(\hat{\boldsymbol{\mu}}),$$

$$A_d(\kappa) = \frac{I_{d/2}}{I_{d/2-1}}, \quad \bar{R} = \frac{||\sum_{i=1}^n\boldsymbol{w}_i||}{n},$$

$$\hat{\kappa} = A_d^{-1}(\bar{R}) \approx \frac{\bar{R}(d-\bar{R}^2)}{1-\bar{R}^2}.$$

where both the derivation and approximation for the MLE estimates are derived in Banerjee et al. (2005). Let $\mathcal{D} = \{\phi_i\}_{i=1}^n$ be the dataset. The log likelihood is then

$$\mathcal{L}(\boldsymbol{\theta}, \kappa|\boldsymbol{\phi}) = \kappa \boldsymbol{w}(\boldsymbol{\phi})^T \boldsymbol{\mu}(\boldsymbol{\theta}) - \log Z(\kappa) + \log \left|\frac{\partial \boldsymbol{w}}{\partial \boldsymbol{\phi}}\right|, \qquad \mathcal{L}(\boldsymbol{\theta}, \kappa|\mathcal{D}) = \sum_{i=1}^n \mathcal{L}(\boldsymbol{\theta}, \kappa|\boldsymbol{\phi}_i).$$

## C  PARTIAL DERIVATIVE CALULATIONS (VMF LIKELIHOOD)

We first show the following result, which is useful for the full derivation. For $k \leq j$

$$\frac{\partial}{\partial \theta_k} \mu_j(\boldsymbol{\theta}) = \frac{\partial}{\partial \theta_k} \left((1 - \delta_{kd}) \cos \theta_k + \delta_{kd}\right) \prod_{i=1}^{j-1} \sin \theta_i$$

$$= \left((1 - \delta_{kj})\frac{\cos \theta_k}{\sin \theta_k} - \delta_{kj}\frac{\sin \theta_k}{\cos \theta_k}\right) \left((1 - \delta_{kd}) \cos \theta_k + \delta_{kd}\right) \prod_{i=1}^{i-1} \sin \theta_i$$

$$= ((1 - \delta_{kj}) \cot \theta_k - \delta_{kj} \tan \theta_k) \mu_j(\boldsymbol{\theta}),$$

where the second line comes from the fact that $\sin \theta_k$ (or $\cos \theta_k$) gets transformed into a $\cos \theta_k$ (or $-\sin \theta_k$), and thus we can revert to the original definition of $\mu_j$ by multiplying with a $\cot \theta_k$ (or $-\tan \theta_k$). If $k > j$, this derivative is 0. Thus, for a single data point $\boldsymbol{w}(\boldsymbol{\phi})$

$$\frac{\partial}{\partial \theta_k} \mathcal{L}(\boldsymbol{\theta}, \kappa|\boldsymbol{\phi}) = \frac{\partial}{\partial \theta_k} \kappa \boldsymbol{w}(\boldsymbol{\phi})^\top \boldsymbol{\mu}(\boldsymbol{\theta}) - \frac{\partial}{\partial \theta_k} \log Z(\kappa)$$

$$= \kappa \boldsymbol{w}(\boldsymbol{\phi})^\top \frac{\partial \boldsymbol{\mu}(\boldsymbol{\theta})}{\partial \theta_k}$$

$$= \kappa \sum_{j=k}^d w_j(\boldsymbol{\phi}) \mu_j(\boldsymbol{\theta})((1 - \delta_{kj}) \cot \theta_k - \delta_{kj} \tan \theta_k),$$

where the sum starts from $k$, as for $j < k$, the derivative is zero.

The derivative with respect to $\kappa$ is derived as follows

$$\frac{\partial}{\partial \kappa} \mathcal{L}(\boldsymbol{\theta}, \kappa|\boldsymbol{\phi}) = \frac{\partial}{\partial \kappa} \kappa \boldsymbol{w}(\boldsymbol{\phi})^\top \boldsymbol{\mu}(\boldsymbol{\theta}) - \frac{\partial}{\partial \kappa} \log Z(\kappa)$$

$$= \boldsymbol{w}(\boldsymbol{\phi})^\top \boldsymbol{\mu}(\boldsymbol{\theta}) - \frac{I_{\frac{d}{2}}(\kappa)}{I_{\frac{d}{2}-1}(\kappa)},$$

where the derivative of the second term is a known result.

We next focus on second order derivatives

$$\frac{\partial^2}{\partial \theta_k^2} \mu_j(\boldsymbol{\theta}) = \frac{\partial^2}{\partial \theta_k^2} \left((1 - \delta_{id}) \cos \theta_i + \delta_{id}\right) \prod_{i=1}^{j-1} \sin \theta_i.$$

Unless this derivative is zero, we notice that we take the derivative $\frac{\partial^2 \cos \theta_k}{\partial \theta_k^2}$ or $\frac{\partial^2 \sin \theta_k}{\partial \theta_k^2}$, both of which result in the negative of the original function. Thus

$$\frac{\partial^2}{\partial \theta_k^2} \mathcal{L}(\boldsymbol{\theta}, \kappa|\boldsymbol{\phi}) = -\kappa \sum_{j=k}^d w_j(\boldsymbol{\phi}) \mu_j(\boldsymbol{\theta}).$$

The below result is given

$$\frac{\partial^2}{\partial \kappa^2} \mathcal{L}(\boldsymbol{\theta}, \kappa|\boldsymbol{\phi}) = \frac{\partial}{\partial \kappa}\left(-\frac{I_{\frac{d}{2}}(\kappa)}{I_{\frac{d}{2}-1}(\kappa)}\right)$$

$$= \frac{I_{\frac{d}{2}}(\kappa)(I_{\frac{d}{2}-2}(\kappa) + I_{\frac{d}{2}}(\kappa)) - I_{\frac{d}{2}-1}(\kappa)(I_{\frac{d}{2}-1}(\kappa) + I_{\frac{d}{2}+1}(\kappa))}{2 I_{\frac{d}{2}-1}(\kappa)^2}.$$

Next, we show that the second order mixed derivatives are a constant (with respect to $\phi$) times $\partial \mathcal{L}(\boldsymbol{\theta}, \kappa | \phi) / \partial \theta_k$, i.e.

$$
\begin{aligned}
\frac{\partial^2 \mathcal{L}(\boldsymbol{\theta}, \kappa | \phi)}{\partial \kappa \partial \theta_k} &= \frac{\partial^2}{\partial \kappa \partial \theta_k} \kappa \boldsymbol{w}(\phi)^\top \boldsymbol{\mu}(\boldsymbol{\theta}) - \frac{\partial}{\partial \kappa \partial \theta_k} \log Z(\kappa) \\
&= \boldsymbol{w}(\phi)^\top \frac{\partial \boldsymbol{\mu}(\boldsymbol{\theta})}{\partial \theta_k} \\
&= \kappa^{-1} \frac{\partial \mathcal{L}(\boldsymbol{\theta}, \kappa | \phi)}{\partial \theta_k},
\end{aligned}
$$

Evaluated at the MLE by definition $\kappa^{-1} \frac{\partial \mathcal{L}(\boldsymbol{\theta}, \kappa | \mathcal{D})}{\partial \theta_k} \Big|_{\boldsymbol{\theta} = \hat{\boldsymbol{\theta}}, \kappa = \hat{\kappa}} = 0$

Assuming $l < k$ (Hessian is symmetric)

$$
\begin{aligned}
\frac{\partial \mathcal{L}(\boldsymbol{\theta}, \kappa | \phi)}{\partial \theta_k \partial \theta_l} &= \kappa \boldsymbol{w}(\phi)^\top \frac{\partial \boldsymbol{\mu}(\boldsymbol{\theta})}{\partial \theta_k \partial \theta_l} \\
&= \kappa \sum_{j=\max(k,l)}^d w_j(\phi) \mu_j(\boldsymbol{\theta}) ((1 - \delta_{kj}) \cot \theta_k - \delta_{kj} \tan \theta_k)((1 - \delta_{lj}) \cot \theta_l - \delta_{lj} \tan \theta_l) \\
&= \kappa \sum_{j=k}^d w_j(\phi) \mu_j(\boldsymbol{\theta}) \cot \theta_l ((1 - \delta_{kj}) \cot \theta_k - \delta_{kj} \tan \theta_k) \\
&= \cot \theta_l \kappa \sum_{j=l}^d w_j(\phi) \mu_j(\boldsymbol{\theta}) ((1 - \delta_{kj}) \cot \theta_k - \delta_{kj} \tan \theta_k) \\
&= \cot \theta_l \frac{\partial \mathcal{L}(\boldsymbol{\theta}, \kappa | \phi)}{\partial \theta_k}.
\end{aligned}
$$

Which is also zero evaluated at the MLE $\cot \theta_l \frac{\partial \mathcal{L}(\boldsymbol{\theta}, \kappa | \mathcal{D})}{\partial \theta_k} \Big|_{\boldsymbol{\theta} = \hat{\boldsymbol{\theta}}, \kappa = \hat{\kappa}} = 0$.

## D  PARTIAL DERIVATIVES CALCULATION (GAUSSIAN LIKELIHOOD)

The partial derivatives for the diagonal Gaussian likelihood are (we take derivatives with respect to precision $\lambda_k^2 = 1/\sigma_k^2$)

$$
\begin{aligned}
\frac{\partial}{\partial \mu_k} \mathcal{L}(\boldsymbol{\mu}, \boldsymbol{\lambda} | \boldsymbol{w}) &= \sum_{i=1}^n \lambda_k^2 \left( x_k^{(i)} - \mu_k \right), \\
\frac{\partial^2}{\partial \mu_k^2} \mathcal{L}(\boldsymbol{\mu}, \boldsymbol{\lambda} | \boldsymbol{w}) &= -n \lambda_k^2, \\
\frac{\partial}{\partial \lambda_k^2} \mathcal{L}(\boldsymbol{\mu}, \boldsymbol{\lambda} | \boldsymbol{w}) &= \frac{n}{2\lambda_k^2} - \frac{1}{2} \sum_{i=1}^n \left( x_k^{(i)} - \mu_k \right)^2, \\
\frac{\partial^2}{\partial (\lambda_k^2)^2} \mathcal{L}(\boldsymbol{\mu}, \boldsymbol{\lambda} | \boldsymbol{w}) &= -\frac{n}{2\lambda_k^4}, \\
\frac{\partial^2}{\partial \lambda_k^2 \partial \mu_k} \mathcal{L}(\boldsymbol{\mu}, \boldsymbol{\lambda} | \boldsymbol{w}) &= \sum_{i=1}^n \left( x_k^{(i)} - \mu_k \right).
\end{aligned}
$$

Evaluating at the MLE we get

$$
\begin{aligned}
\frac{\partial^2}{\partial \mu_k^2} \mathcal{L}(\boldsymbol{\mu}, \boldsymbol{\lambda} | \mathcal{D}) \Big|_{\boldsymbol{\mu} = \hat{\boldsymbol{\mu}}, \boldsymbol{\lambda} = \hat{\boldsymbol{\lambda}}} &= -n \hat{\lambda}_k^2 = -\frac{n}{\hat{\sigma}_k^2}, \\
\frac{\partial^2}{\partial (\lambda_k^2)^2} \mathcal{L}(\boldsymbol{\mu}, \boldsymbol{\lambda} | \mathcal{D}) \Big|_{\boldsymbol{\mu} = \hat{\boldsymbol{\mu}}, \boldsymbol{\lambda} = \hat{\boldsymbol{\lambda}}} &= -\frac{n}{2\hat{\lambda}_k^4} = -\frac{n\hat{\sigma}_k^4}{2}, \\
\frac{\partial^2}{\partial \lambda_k^2 \partial \mu_k} \mathcal{L}(\boldsymbol{\mu}, \boldsymbol{\lambda} | \mathcal{D}) \Big|_{\boldsymbol{\mu} = \hat{\boldsymbol{\mu}}, \boldsymbol{\lambda} = \hat{\boldsymbol{\lambda}}} &= \sum_{i=1}^n \left( x_k^{(i)} - \hat{\mu}_k \right) = 0.
\end{aligned}
$$

Substituting these derivatives into the definition of $\hat{\boldsymbol{\mathcal{I}}}$ and $\hat{\boldsymbol{J}}$ we get

$$\hat{\boldsymbol{\mathcal{I}}}_{\mu_k, \mu_k} = \hat{\lambda}_k^2, \quad \hat{\boldsymbol{\mathcal{I}}}_{\lambda_k^2, \lambda_k^2} = -4\hat{\lambda}_k^{-4} + \frac{(\hat{\boldsymbol{\mu}}_4)_k}{4}$$

$$\hat{\boldsymbol{J}}_{\mu_k, \mu_k} = -\hat{\lambda}_k^2, \quad \hat{\boldsymbol{J}}_{\lambda_k^2, \lambda_k^2} = -\frac{1}{2}\hat{\lambda}_k^{-4}$$

Finally, computing the model complexity penalty we get

$$\mathrm{tr}(\hat{\boldsymbol{\mathcal{I}}}\hat{\boldsymbol{J}}^{-1}) = \frac{1}{2}\left(d + \sum_{i=1}^{d} \frac{(\hat{\boldsymbol{\mu}}_4)_i}{\hat{\sigma}_i^4}\right) = \frac{d}{2} + \sum_{i=1}^{d} \frac{\hat{\kappa}_i}{2}.$$

# E  BAYES FACTOR AND BAYESIAN INFORMATION CRITERION

We first define the Bayes Factor for the Gaussian likelihood with parameters $(\boldsymbol{\mu}, \boldsymbol{\Lambda} = \boldsymbol{\Sigma}^{-1})$. We assume a Wishart prior

$$p(\boldsymbol{\mu}, \boldsymbol{\Lambda}) = \mathcal{N}\left(\boldsymbol{\mu}|\boldsymbol{\mu}_0, (\kappa_0\boldsymbol{\Lambda})^{-1}\right)\mathcal{W}_i(\boldsymbol{\Lambda}|\nu_0, \boldsymbol{T}_0^{-1}),$$

which yields the following Normal-Wishart posterior (Murphy, 2007)

$$p(\boldsymbol{\mu}, \boldsymbol{\Lambda}|\mathcal{D}) = \mathcal{N}\left(\boldsymbol{\mu}|\boldsymbol{\mu}_n, (\kappa_n\boldsymbol{\Lambda})^{-1}\right)\mathcal{W}_i(\boldsymbol{\Lambda}|\nu_n, \boldsymbol{T}_n^{-1}).$$

The posterior parameters are

$$\nu_n = \nu_0 + n,$$
$$\kappa_n = \kappa_0 + n,$$
$$\boldsymbol{\mu}_n = \frac{\kappa_0\boldsymbol{\mu}_0 + n\bar{\mathbf{w}}}{\kappa_n},$$
$$\boldsymbol{S} = \sum_{k=1}^{n}\left(\boldsymbol{w}_k - \bar{\boldsymbol{w}}\right)\left(\boldsymbol{w}_k - \bar{\boldsymbol{w}}\right)^{\top},$$
$$\boldsymbol{T}_n = \boldsymbol{S} + \boldsymbol{T}_0 + \frac{n\kappa_0}{2\kappa_n}\left(\bar{\mathbf{w}} - \boldsymbol{\mu}_0\right)\left(\bar{\mathbf{w}} - \boldsymbol{\mu}_0\right)^{\top}.$$

The evidence (Murphy, 2007) is

$$p(\mathcal{D}) = \frac{1}{\pi^{nd/2}}\left(\frac{\kappa_0}{\kappa_n}\right)^{\frac{d}{2}}\frac{|\boldsymbol{T}_0|^{\frac{\nu_0}{2}}}{|\boldsymbol{T}_n|^{\frac{\nu_n}{2}}}\frac{\Gamma_d(\nu_n/2)}{\Gamma_d(\nu_0/2)}.$$

Using $p(\mathcal{D}_1, \mathcal{D}_2|\mathcal{M}_1) = p(\mathcal{D}_1 \oplus \mathcal{D}_2|\mathcal{M}_1)$ we compute the Bayes factor for $\mathcal{M}_1, \mathcal{M}_2$ in closed form

$$\mathbf{sim}(\mathcal{D}_1, \mathcal{D}_2) = \left(\frac{\kappa_n\kappa_m}{\kappa_0\kappa_l}\right)^{\frac{d}{2}}\frac{|\boldsymbol{T}_n|^{\frac{\nu_n}{2}}|\boldsymbol{T}_m|^{\frac{\nu_m}{2}}}{|\boldsymbol{T}_l|^{\frac{\nu_l}{2}}|\boldsymbol{T}_0|^{\frac{\nu_0}{2}}}\frac{\Gamma_d\left(\frac{\nu_l}{2}\right)\Gamma_d\left(\frac{\nu_l}{2}\right)}{\Gamma_d\left(\frac{\nu_n}{2}\right)\Gamma_d\left(\frac{\nu_m}{2}\right)}. \tag{3}$$

where $|\mathcal{D}_1| = n$, $|\mathcal{S}_2| = m$ and $|\mathcal{D}_1 \oplus \mathcal{D}_2| = m + n = l$.

The BIC is defined as

$$\mathrm{BIC}(\mathcal{D}, \mathcal{M}) = -2\mathcal{L}(\hat{\boldsymbol{\theta}}|\mathcal{D}, \mathcal{M}) + k\log n \approx -p(\mathcal{D}|\mathcal{M}),$$

and acts as a direct approximation to the model evidence (Schwarz et al., 1978). Thus, the similarity under the BIC is

$$\mathbf{sim}(\mathcal{D}_1, \mathcal{D}_2) = 2\left(\mathcal{L}(\hat{\boldsymbol{\theta}}_{1,2}|\mathcal{M}_1) - \mathcal{L}(\hat{\boldsymbol{\theta}}_1|\mathcal{M}_2) - \mathcal{L}(\hat{\boldsymbol{\theta}}_2|\mathcal{M}_2)\right) - k\log\frac{n+m}{nm}, \tag{4}$$

|               | AIC        | Bayes Factor | BIC    |
| ------------- | ---------- | ------------ | ------ |
| STS-12        | **0.6031** | 0.5183       | 0.5009 |
| STS-13 (-SMT) | 0.6132     | **0.6297**   | 0.6011 |
| STS-14        | **0.6445** | 0.6428       | 0.5926 |
| STS-15        | 0.7171     | **0.7429**   | 0.6625 |
| STS-16        | **0.7346** | 0.6328       | 0.5826 |

Table 4: Spearman correlations using GloVe embeddings and Gaussian likelihood. The AIC and BIC use a diagonal covariance matrix, while the Bayes Factor uses a full covariance matrix.

where $n, m$ are defined as above.

Equations 3 and 4 represents our similarity score under a Gaussian likelihood, for the Bayes Factor and BIC respectively.

Table E compares BIC and the Bayes Factor using a Gaussian likelihood to the approach presented in the paper. We see that while these approaches are competitive on STS-13, STS-14 and STS-15, they both give severely worse results on STS-12 and STS-16, with more than 0.08 difference. This motivates our choice to do a penalised likelihood ratio test instead of doing full Bayesian inference of the evidences.

## F  TIC ROBUSTNESS

In this section, we compare the TIC and AIC on the two likelihoods described in the main text. Table F presents the results of that comparison. As we can see, with each word embedding the Gaussian AIC correction outperforms the TIC correction on average. Looking at Figure 3, it becomes apparent why — the more parameters a Gaussian has, the more dependent its correction is on the number of words in the sentence. This is reminiscent of the linear scaling with number of words in the BIC penalty discussed in Appendix E, which was shown to perform badly. On the other hand, looking at Figure 4, we see that the TIC for the vMF distribution has very low variance, and is generally not dependent on the number of word embeddings in the word group. This gives intuition why the AIC and TIC for the vMF give very similar results.

| Embedding | Method   | STS12      | STS13      | STS14      | STS15      | STS16      | Average    |
| --------- | -------- | ---------- | ---------- | ---------- | ---------- | ---------- | ---------- |
| FastText  | vMF+TIC  | **0.5578** | **0.5532** | **0.5897** | **0.6660** | 0.6596     | **0.6023** |
|           | vMF+AIC  | 0.5443     | 0.5445     | 0.5853     | 0.6636     | **0.6654** | 0.5966     |
|           | Diag+TIC | 0.5883     | **0.6553** | 0.6678     | 0.7197     | 0.7052     | 0.6626     |
|           | Diag+AIC | **0.6193** | 0.6335     | **0.6721** | **0.7328** | **0.7518** | **0.6764** |
| GloVe     | vMF+TIC  | 0.5516     | **0.5873** | **0.5865** | **0.6605** | 0.6083     | **0.5977** |
|           | vMF+AIC  | **0.5541** | 0.5739     | 0.5824     | 0.6576     | **0.6599** | **0.5997** |
|           | Diag+TIC | 0.5802     | **0.6420** | **0.6535** | **0.7179** | 0.6966     | 0.6534     |
|           | Diag+AIC | **0.6031** | 0.6132     | 0.6445     | **0.7171** | **0.7346** | **0.6564** |
| Word2Vec  | vMF+TIC  | 0.5161     | 0.4997     | 0.5661     | 0.6477     | 0.5984     | 0.5683     |
|           | vMF+AIC  | **0.5176** | **0.5045** | **0.5696** | **0.6495** | **0.6072** | **0.5716** |
|           | Diag+TIC | 0.5366     | **0.5932** | 0.6258     | 0.6814     | 0.6215     | 0.6127     |
|           | Diag+AIC | **0.5630** | 0.5799     | **0.6291** | **0.6951** | **0.6701** | **0.6265** |

Table 5: Comparison of Spearman correlations on the STS datasets between the TIC and AIC corrections for the diagonal covariance Gaussian and vMF likelihood functions.

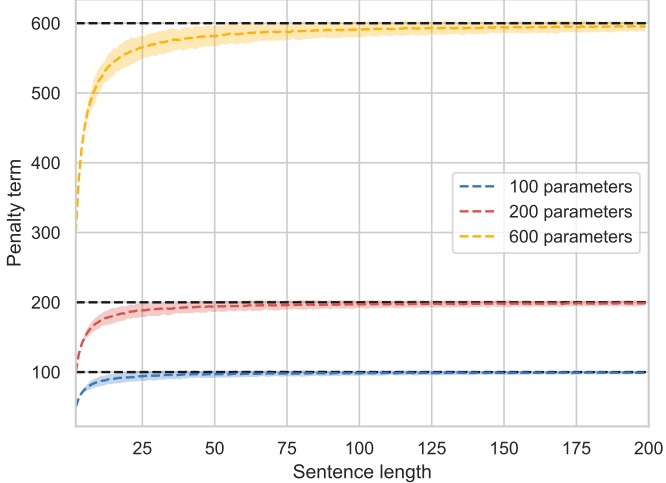

Figure 3: TIC Correction penalty for varying sample sizes, samples generated from standardised normal $\mathcal{N}(\mathbf{0}, \mathbb{I}_d)$.

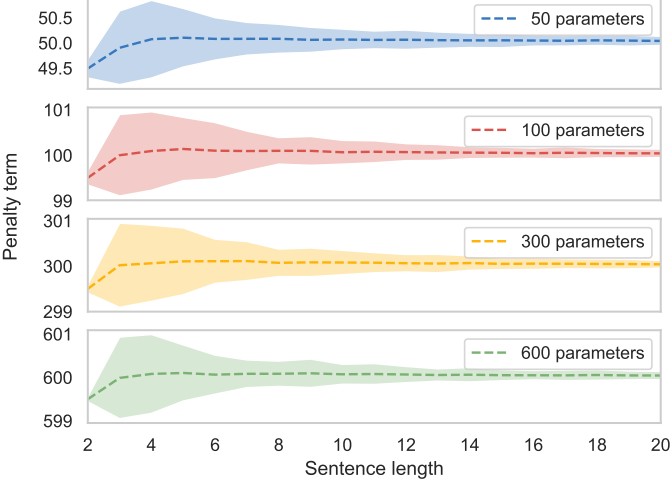

Figure 4: TIC Correction penalty for varying sample sizes, samples generated from Uniform distribution on the unit hypersphere $U(\mathbb{S}_{d-1})$.

