# OpenReview forum: "Model Comparison for Semantic Grouping"
_ICLR.cc/2019/Conference_

### Official Review · AnonReviewer1 · 2018-11-03
**Interesting paper but lacking both context and comprehensive analyses**

**Rating:** 5
**Confidence:** 1

**Review:**

Main contribution: devising and evaluating a theoretically-sound algorithm for quantifying the semantic similarity between two pieces of text (e.g., two sentences), given pre-trained word embeddings (glove).

Clarity:
The paper is generally well-written, but I would have liked to see more details regarding the motivation for the work, description of the prior work and discussion of the results. As an example, I could not understand what were the differences between the online and offline settings, with only a reference to the (Arora et al. 2016) paper that does not contain neither "online" nor "offline". The mathematical derivations are detailed, which is nice.

Originality:
The work looks original. It proposes a method for quantifying semantic similarity that does not rely on cosine similarity.

Significance:
I should start by saying I am not a great reviewer for this paper. I am not familiar with the STS dataset and don't have the mathematical background to fully understand the author's algorithm.
I like to see theoretical work in a field that desperately needs some, but overall I feel the paper could do a much better job at explaining the motivation behind the work, which is limited to "cosine similarity [...] is not backed by a solid theoretical foundation".
I am not convinced of the practicality of the algorithm either: the algorithm seems to improve slightly over the compared approaches (and it is unclear if the differences are significant), and only in some settings. The approach needs to remove stop-words, which is reminiscent of good old feature engineering. Finally, the paper claims better average time complexity than some other methods, but discussing whether the algorithm is faster for common ranges of d (the word embedding dimension) would also have been interesting.

---

> ### Author Response · Authors · 2018-11-13
> **Clarifications and updated results (Gaussian likelihood function)**
>
> We would like to thank the reviewer for their comments. We have tried to address their concerns below.
>
> "I could not understand what were the differences between the online and offline settings..."
>
> We apologise for the lack of definition of these terms in the paper. This will be remedied in the next version. The difference between an online and offline setting is whether one has access to the entire dataset on which the methods will be evaluated at once. Information retrieval is an example setting in which one cannot perform the PCA on the query dataset as seen in (Arora et al. 2017), since new queries will arrive in an online fashion.
>
> "...the paper could do a much better job at explaining the motivation behind the work..."
>
> Similarity measures are often not theoretically justified - for example, cosine similarity is preferred to dot product similarity based purely on empirical results. It is difficult for a practitioner to utilize word vectors efficiently if the underlying assumptions in the similarity measure are not well understood. Our framework addresses these issues by explicitly deriving the similarity through the likelihood of the chosen generative process, instead of empirically motivating the similarity measure. Via designing the likelihood the practitioner can encode suitable assumptions and constraints that may be favourable to the application of interest. Furthermore, this framework proposes a new research direction that could help understand semantic similarity, in which practitioners can study suitable distributions and see how these perform. We will elaborate on this further in the updated version of the manuscript.
>
> The second motivation of this work is to derive a simple but competitive similarity measure that would perform well in online settings (as defined above). Online settings are both practical and key to use-cases that involve information retrieval in dialogue systems. For example, in a chat-bot application new queries will arrive in an online fashion and methods such as SIF+PCA will not perform as strongly as they do on STS. This is because the method itself (in this case the PCA part) was fitted on the reported test set, which will not be available a priori in online settings.
>
> "I am not convinced of the practicality of the algorithm..."
>
> We were unsure of what the reviewer meant by practicality of the algorithm. The presented algorithm requires no more than 30 lines of code to implement once the derivatives for the chosen likelihood have been calculated. Furthermore, the derivatives can be computed automatically with frameworks such as autograd, TensorFlow, PyTorch, and others.
>
> Below we provide the results when using a multivariate Gaussian distribution with diagonal covariance as a likelihood.
>
> +--------------+-------------+-----------+----------+----------+----------+----------+----------+
> |                  |   Method | STS12  | STS13  | STS14  | STS15  | STS16| W. A.* |
> +--------------+-------------+-----------+----------+----------+----------+----------+----------+
> | fasttext   | Ours        | 0.6193 | 0.6335 | 0.6721 | 0.7328 | 0.7518| 0.6765|
> |                  | SIF+PCA  | 0.5893 | 0.7121 | 0.6790 | 0.7498 | 0.7142| 0.6810|
> +--------------+-------------+-----------+----------+----------+----------+----------+----------+
> | glove       | Ours        | 0.6031 | 0.6132 | 0.6445 | 0.7171 | 0.7346| 0.6564|
> |                  | SIF+PCA  | 0.5681 | 0.6844 | 0.6546 | 0.7166 | 0.6931| 0.6552|
> +--------------+-------------+-----------+----------+----------+----------+----------+----------+
> | word2vec| Ours       | 0.5630 | 0.5799 | 0.6291 | 0.6951 | 0.6701| 0.6265|
> |                  | SIF+PCA  | 0.5324 | 0.6486 | 0.6510 | 0.7031| 0.6609| 0.6347|
> +--------------+-------------+-----------+----------+----------+----------+----------+----------+
> * W. A. stands for weighted average
>
> As can be seen from the table, the results for this likelihood are on par with the ones presented in (Arora et al. 2017).
>
> "The approach needs to remove stop-words, which is reminiscent of good old feature engineering."
>
> [EDIT] We have decided to report only results of experiments without any stopword removal. As the reviewers suggested, stopword removal benefits some approaches more than it does others. This decision also heavily reduces the clutter of the paper and the amount of care that needs to be taken to reproduce our results.
>
> "...discussing whether the algorithm is faster for common ranges of d (the word embedding dimension)..."
>
> We appreciate the suggestion of grounding the complexity analysis with values for N and D in the ranges experienced in the STS dataset. We will provide an analysis with the next version of the manuscript.

---

### Official Review · AnonReviewer2 · 2018-11-03
**Interesting model, but would like to see some more motivation**

**Rating:** 5
**Confidence:** 3

**Review:**

The authors propose a probabilistic model for computing the sentence similarity between two sets of representations in an online fashion (that is, they do not need to see the entire dataset at once as SIF does when using PCA). They evaluate on the STS tasks and outperform competitive baselines like WMD, averaging embeddings, and SIF (without PCA), but they have worse performance that SIF + PCA.

The paper is clearly written and their model is carefully laid out along with their derivation. My concern with this paper however, is that I feel the paper lacks a motivation, was it derive an online similarity metric that outperforms SIF(without PCA)?

A few experimental questions/comments:

What happens to all methods when stop words are not removed? How far does performance fall? I think one reason it might fall (in addition to the reasons given in the paper) is that all vectors are set to have the same norm. For STS tasks, often the norms of these vectors are reduced during training which lessens their influence. What mechanism was used to identify the stop words and does removing these help the other methods (I know in the paper, stop words were removed in the baseline, did this unilaterally improve performance for these methods)?

Overall I do like the paper, however I do find the results to be lackluster. There are many papers on combining word embeddings trained in various ways that have much stronger numbers on STS, but these methods won't be effective with this type of similarity (namely because embeddings must have unit norm in their model). Therefore, I think the paper needs some more motivation and experimental evidence of its superiority over related methods like SIF+PCA in order for it to be accepted.

PROS
- Probabilistic model with clear design assumptions from which a similarity metric can be derived.
- Derived similarity metric doesn't require knowledge of the entire dataset (in comparison to SIF + PCA)

CONS
- Performance seems to be slightly better than SIF, WMD, and averaging word embeddings, but below that of SIF + PCA
- Unclear motivation for the model, was it derive an online similarity metric that outperforms SIF(without PCA)?
- Requires the removal of stop words, but doesn't state how these were defined. Minor point, but tuning this could be enough to cause the improvement over related methods.

---

> ### Author Response · Authors · 2018-11-13
> **Clarifications and updated results (Gaussian likelihood function)**
>
> We would like to thank the reviewer for their in-depth feedback. Below, we present preliminary results, as well as clarifications on the conceptual questions that were posed.
>
> "My concern with this paper however, is that I feel the paper lacks a motivation..."
>
> The main focus of this paper is the introduction of a framework that allows for clear assumptions to be made about the distribution of word vectors in a sentence via a choice of likelihood and for these assumptions to be tested on the STS benchmark - any likelihood will fit into this general framework and produce a similarity measure. This allows practitioners to design likelihoods that encode suitable properties for their application.
>
> The more practical motivation for this paper is that the online* setting is key for many real-world use-cases such as information retrieval for dialogue systems (i.e. chatbots) where new queries will arrive in an online fashion and methods like SIF+PCA will not be as applicable as they are in the STS task. Whilst the method is derived as an online method it can be used in applications that have offline components, and we have shown it remains competitive to offline methods such as SIF+PCA (see the next section of this response).
>
> *We thank the reviewer for helpfully clarifying the definitions of an "online" and "offline" setting; we will include this definition in the next version of the manuscript.
>
> "...namely because embeddings must have unit norm in their model."
>
> The reviewer has helpfully pointed out an implicit assumption that we made - namely, we assumed that the magnitude of word embedding was noise rather than useful information. To test this assumption, we are running experiments with a multivariate Gaussian likelihood with diagonal covariance. This does not require unit norming the vectors and a set of preliminary results are presented below.
>
> +--------------+-------------+-----------+----------+----------+----------+----------+----------+
> |                  |   Method | STS12  | STS13  | STS14  | STS15  | STS16| W. A.* |
> +--------------+-------------+-----------+----------+----------+----------+----------+----------+
> | fasttext   | Ours        | 0.6193 | 0.6335 | 0.6721 | 0.7328 | 0.7518| 0.6765|
> |                  | SIF+PCA  | 0.5893 | 0.7121 | 0.6790 | 0.7498 | 0.7142| 0.6810|
> +--------------+-------------+-----------+----------+----------+----------+----------+----------+
> | glove       | Ours        | 0.6031 | 0.6132 | 0.6445 | 0.7171 | 0.7346| 0.6564|
> |                  | SIF+PCA  | 0.5681 | 0.6844 | 0.6546 | 0.7166 | 0.6931| 0.6552|
> +--------------+-------------+-----------+----------+----------+----------+----------+----------+
> | word2vec| Ours       | 0.5630 | 0.5799 | 0.6291 | 0.6951 | 0.6701| 0.6265|
> |                  | SIF+PCA  | 0.5324 | 0.6486 | 0.6510 | 0.7031| 0.6609| 0.6347|
> +--------------+-------------+-----------+----------+----------+----------+----------+----------+
> * W.A. stands for weighted average
>
> "...I do find the results to be lackluster."
>
> As we can see, the Gaussian distribution seems to be a better fit than the vMF one, matching SIF+PCA on the three word embeddings we tested on. We hope this addresses the concern of the reviewer that methods which depend on embedding magnitude won't be applicable with this framework. We will include a more thorough set of results in the next version of the manuscript.
>
> "What mechanism was used to identify the stop words and does removing these help the other methods..."
> "What happens to all methods when stop words are not removed?"
>
> [EDIT] We have decided to report only results of experiments without any stopword removal. As the reviewers suggested, stopword removal heavily benefited the vMF likelihood, while the newly introduced Gaussian likelihood proves to be more robust. This decision also heavily reduces the clutter of the paper and the amount of care that needs to be taken to reproduce our results.

---

### Official Review · AnonReviewer3 · 2018-11-08
**Interesting idea but somewhat incomplete study**

**Rating:** 5
**Confidence:** 3

**Review:**

The paper proposes a Bayesian model comparison based approach for quantifying the semantic similarity between two groups of embeddings (e.g., two sentences). In particular, it proposes to use the difference between the probability that the two groups are from the same model and the probability that they are from different models.

While the approach looks interesting, I have a few concerns:
-- Using the Bayesian model comparison framework seems to be an interesting idea. However, what are the advantages compared to widely used learned models (say, a learned CNN that takes as input two sentences and outputs the similarity score)? The latter can fit the ground-truth labels given by humans, while it's unclear the model comparison leads to good correlation with human judgments. Some discussion should be provided.
-- The von Mises-Fisher Likelihood is a very simplified model of actual text data. Have you considered using other models? In particular, more sophisticated ones may lead to better performance.
-- Different information criteria can be plugged in. Are there comparisons?
-- The experiments are just too simple and incomplete to make reasonable conclusions. For example, it seems compared to SIF there is not much advantage even in the online setting.

---

> ### Author Response · Authors · 2018-11-13
> **Clarifications and updated results (Gaussian likelihood function) [Part 2]**
>
> "The latter can fit the ground-truth labels given by humans, while it's unclear the model comparison leads to good correlation with human judgments. Some discussion should be provided."
>
> STS provides a test set in order to evaluate how the methods correlate with human scores, which we have used to benchmark our proposed models. That is, performing well on the test set suggests there's a correlation between human judgment and the model's prediction of similarity score. We will clarify this in the new manuscript.
>
> We will discuss the relation of our approach to the one presented in Equation (9) (Tversky's contrast model) of [6] - a work that analyses what a good similarity is from a cognitive science perspective.
>
> [6] J.B. Tenenbaum, and T.L. Griffiths, 2001. Generalization, similarity, and Bayesian inference. Behavioral and brain sciences.
>
> "Have you considered using other models? In particular, more sophisticated ones may lead to better performance."
>
> The motivation for this paper is to introduce a framework in which different probabilistic models can be assessed on the STS task. This is done such that a practitioner, through specifying the likelihood function, can encode suitable assumptions and constraints that may be favourable to the application of interest. The primary goal of this work is not to find the most accurate model, however we hope that this framework could be a stepping stone in using more complex and accurate generative models of text to asses semantic similarity. In the next draft of the paper we will add two different likelihoods which allow for non-unit normed vectors, unlike the vMF distribution.
>
> "The experiments are just too simple and incomplete to make reasonable conclusions."
>
> We are unsure if the reviewer has concerns about the STS task in particular, or the variety of experiments ran.
>
> To address the former, we provide our argument for why we think STS is an adequate task to assess performance on. Our focus is on the setting where one has word level embeddings that contain semantic information about individual words, but no labelled corpus with examples of similar and dissimilar pairs (an unsupervised setting). Furthermore, we assume sentences arrive in an 'online' fashion meaning that we don't have access to the whole sentence corpus a priori. An example use-case like this is a chat-bot application.
>
> To address the latter, we will extend our experiments by considering other word vectors usually used to assess performance on STS such as fasttext and word2vec. We will also include experimental results using other likelihoods and information criteria within our framework. Below we provide a preliminary set of results using a Gaussian likelihood with a diagonal covariance matrix.
>
> +--------------+-------------+-----------+----------+----------+----------+----------+----------+
> |                  |   Method | STS12  | STS13  | STS14  | STS15  | STS16| W. A.* |
> +--------------+-------------+-----------+----------+----------+----------+----------+----------+
> | fasttext   | Ours        | 0.6193 | 0.6335 | 0.6721 | 0.7328 | 0.7518| 0.6765|
> |                  | SIF+PCA  | 0.5893 | 0.7121 | 0.6790 | 0.7498 | 0.7142| 0.6810|
> +--------------+-------------+-----------+----------+----------+----------+----------+----------+
> | glove       | Ours        | 0.6031 | 0.6132 | 0.6445 | 0.7171 | 0.7346| 0.6564|
> |                  | SIF+PCA  | 0.5681 | 0.6844 | 0.6546 | 0.7166 | 0.6931| 0.6552|
> +--------------+-------------+-----------+----------+----------+----------+----------+----------+
> | word2vec| Ours       | 0.5630 | 0.5799 | 0.6291 | 0.6951 | 0.6701| 0.6265|
> |                  | SIF+PCA  | 0.5324 | 0.6486 | 0.6510 | 0.7031| 0.6609| 0.6347|
> +--------------+-------------+-----------+----------+----------+----------+----------+----------+
> * W.A. stands for weighted average
>
> Would experiments along these lines address the simplicity concern of the reviewer?

---

> ### Author Response · Authors · 2018-11-13
> **Clarifications and updated results (Gaussian likelihood function) [Part 1]**
>
> We want to thank the reviewer for the suggested directions on motivating this work. We will try to address the main concerns below and will deffer the details for the next version of the manuscript.
>
> "The paper proposes a Bayesian model comparison based approach for quantifying the semantic similarity between two groups of embeddings (e.g., two sentences)."
>
> We would like to clarify a point here that didn't come across clearly enough in our submission. Unlike prior work [1, 2], we are not carrying out Bayesian model comparison - we cover this approach in depth since it is the most relevant prior work to our framework. We carefully review why Bayesian model comparison may not be well suited to this application due to the Bayes Factor's sensitivity to the prior [3, 4]. In order to overcome this we propose model comparison criteria that minimise KL divergence across a candidate set of models. This results in a penalized likelihood ratio test which gives competitive results. We will further clarify this difference in the next version of the manuscript.
>
> We have carried out additional experiments using the Bayes Factor and Bayesian Information Criteria (BIC). The results given by these approaches under-perform significantly compared to the information theoretic based criteria. We have provided empirical [5] and theoretical [3, 4] justifications as to why these two techniques are not well suited for the STS task. We will also add the experimental evidence that both the Bayes Factor under a vague prior and BIC perform poorly on the STS task.
>
> [1] P. Marshall et al. Bayesian evidence as a tool for comparing datasets. Physical Review D, 2006.
> [2] Z. Ghahramani and K. Heller. Bayesian sets. In Advances in neural information processing systems, 2006.
> [3] M. Bartlett.  A comment on D. V. Lindley’s statistical paradox. Biometrika, 1957
> [4] H. Akaike et al. Likelihood of a model and information criteria. Journal of econometrics, 1981
> [5] J. Dziak et al. Sensitivity and specificity of information criteria. The Methodology Center and Department of Statistics, The Pennsylvania State University, 2012.
>
> "What are the advantages compared to widely used learned models (say, a learned CNN that takes as input two sentences and outputs the similarity score)?"
>
> A supervised approach such as the one suggested by the reviewer would definitely be an interesting research direction. It is often argued that generative models, such as the one proposed, are less susceptible to over-fitting on small training sets than discriminative models. Discriminative models are likely to fit noise in small training sets, such as the STS data set, which has in the order of thousands of labeled pairs. For this reason, common competitive approaches in the domain mainly rely on either semi-supervised or unsupervised learning procedures.
>
> Semi-supervised approaches (used in STS) do not use human labelled similarity pairs to train on, but instead train a supervised objective on a different task with plenty of data (such as aligned paraphrases) and then use the learned representations from these as sentence embeddings. The general focus of the STS task is in the unsupervised or low-resource setting.
>
> It may be very costly to obtain a large enough labelled dataset for some of the supervised methods to be able to generalize in domain specific applications. This gives a practical motivation for the unsupervised approaches. We will discuss these comparisons and motivations in more detail in the updated version of the manuscript.

---

### Author Response · Authors · 2018-11-26
**Paper draft updated; Modifications summary**

We would like to again thank the reviewers for their time. Taking their feedback into account, we have implemented the following changes:

- Provided a more thorough motivation for our work in the Introduction and Background sections of the paper.
- Clarified the difference between online and offline scenarios, as well as given examples for each of them.
- Moved most of the mathematical work into the Appendix, in order to improve ease of reading.
- Added an additional section introducing the Gaussian likelihood, to illustrate a second example likelihood under our framework.
- Presented only results with stopwords, to address the concerns of unfair comparison with other methods.
- Conducted more experiments using Word2Vec and FastText embeddings, in addition to the original GloVe embeddings.
- Matched the results in Arora et al. (with principal component removal) more closely using the newly introduced Gaussian likelihood, using purely word embedding information, and no external inverse frequencies weights like in Arora et al.
- Showed how our framework can be used to test assumptions about properties of word embeddings.
- Provided further experiments illustrating the poor performance of the Bayes Factor and BIC in Appendix E.

With thanks,
The authors of Paper680

---

> ### Public Comment · (anonymous) · 2018-12-11
> **uSIF instead of SIF?**
>
> I think using uSIF (Ethayarajh, 2018) would be a better comparison for your method than SIF. uSIF fixes some of the problems with SIF and as a result does much better on the STS tasks, achieving state-of-the-art.
>
> The GloVe version of SIF does better than your approach for STS'12, STS'13, STS'14, and STS'15 (STS'16 is not given in the uSIF paper).
>
> [1] https://github.com/kawine/usif

---

> > ### Author Response · Authors · 2018-12-11
> > **uSIF uses PCA on the test corpus, thus it would replace SIF+PCA , not SIF itself. (Please read algorithm 1 in (Ethayarajh, 2018) ) .**
> >
> > In Ethayarajh, 2018 (lines 12-19 of Algorithm 1) you will notice that the uSIF sentence embedding uses PCA (for removing the common discourse vector) on the STS sentence corpus making it an offline algorithm, thus it should be compared in the offline methods section. These methods require having the sentence corpus a priori in order to function at their best, you can observe a notable performance difference between SIF + PCA and just SIF.  In short, uSIF is related to SIF + PCA (as opposed to just SIF), and is not applicable in online information retrieval cases. Ethayarajh, 2018 does not provide comparisons with an alternate method that works in online settings (not carrying out the discourse vector removal). Thus, in order to run a valid comparison with online methods we would have to modify the implementation provided in Ethayarajh, 2018 and remove the principal component subtraction element from it, as done in Arora et al. 2017 (this amounts to setting m=0 in https://github.com/kawine/usif).  This test would be required to show that the online version of uSIF significantly outperform our method in online scenarios.
> >
> > It is worth noting that in Ethayarajh, 2018 a different correlation metric (Pearson instead of Spearman) is used, and thus the results are not directly comparable (although conclusions can be drawn due to its relative performance to SIF+PCA). Moreover, the intention of this work is not to outperform all existing baselines or to achieve state of the art, but to propose a probabilistic framework for deriving similarity measures. We never claim state of the art in the paper, we just state that our method remains strongly competitive to existing online baselines which still holds true.
> >
> > Finally, Ethayarajh, 2018 was published as a workshop paper in July 2018, which was close to the submission deadline. It is difficult to keep up with every new paper, especially if it was published within 2 months before the ICLR deadline. We were unable to find another STS-related submission in ICLR 2018 that cites this paper, so we hope we have convinced the reviewer that the omission is justified.

---

> > ### Author Response · Authors · 2018-12-12
> > **Ethayarajh, 2018 does not perform a fair comparison to Arora et al., 2016.  (uSIF is not better than SIF)**
> >
> > We thank the reviewer again for their suggestion to look into the uSIF method presented in Ethayarajh, 2018. We have carefully reviewed the paper and reproduced results using the source code that the reviewer kindly provided. However, upon inspection of the source code, we can observe that preprocessing (which is not part of the random walk model proposed) on the sentences and word vectors has been carried out selectively on the uSIF method and not on the baseline compared against - Arora et al., 2016’s SIF.
> >
> > The preprocessing includes:
> > - Removal of all non-alphanumeric characters  (https://github.com/kawine/usif/blob/d5bbaab750da644815ce8d84404b67b5a6c710c9/usif.py#L116 )
> > - Custom tokenization on top of NLTK tokenize (the sentEval framework uses a different Tokeniser,  likewise with Arora et al.  which made it impossible to reproduce these results under sentEval)    (https://github.com/kawine/usif/blob/d5bbaab750da644815ce8d84404b67b5a6c710c9/usif.py#L119-L125 )
> > - L_2 normalization along word vector dimensions within a sentence (https://github.com/kawine/usif/blob/d5bbaab750da644815ce8d84404b67b5a6c710c9/usif.py#L133  ). Please note that this heuristic is not part of the modelling used to derive uSIF and does not directly address the issues that uSIF claims to solve, it is introduced as a way of standardising the variance along each dimension (without clear theoretical motivations as to why).  This preprocessing can and should be applied to SIF and other methods in order to guarantee a fair comparison.
> >
> > In order to fairly compare to SIF in Arora et al., 2016, Ethayarajh, 2018 should have run the SIF baseline under their customised STS_Test framework (https://github.com/kawine/usif/blob/master/usif.py ) and should have applied the same sentence processing techniques to SIF. We show in Table 1 that the huge gain in performance demonstrated in Ethayarajh, 2018 is completely due to the combinations of the text preprocessing and the L_2 normalisation along dimensions. Normalisation alone is responsible for up to an absolute increase of 0.07 (7%) of Pearson correlation in some STS years. The punctuation and symbol filtering is on average responsible for an absolute increase of 0.025-0.03 (2.5% - 3%) in Pearson correlation.
> >
> > Overall, from the table below, we can see that the SIF weighting introduced in Arora et al., 2016 is on-par with uSIF when applying the same text normalisation techniques and the L_2 norm on word vector dimensions. Thus we can conclude that our method would be competitive to uSIF under a fair comparison.
> >
> > +----------------------------+-----------+----------+----------+-----------+
> > | Method                      | STS12  | STS13  | STS14  | STS15  |
> > +----------------------------+-----------+----------+----------+-----------+
> > | uSIF - PCA5 - norm  | 0.6043 | 0.6063 | 0.6700 | 0.6361 |
> > | SIF - PCA   - norm    | 0.6014 | 0.6016 | 0.6646 | 0.6286 |
> > +----------------------------+-----------+----------+----------+-----------+
> > | uSIF - PCA5 + norm  | 0.6272 | 0.6765 | 0.7240 | 0.7260 |
> > | SIF - PCA   + norm    | 0.6275 | 0.6732 | 0.7226 | 0.7215 |
> > +----------------------------+-----------+----------+----------+-----------+
> > | uSIF + PCA5 + norm | 0.6493 | 0.7174 | 0.7439 | 0.7612 |
> > | SIF + PCA   + norm   | 0.6459 | 0.7089 | 0.7366 | 0.7517 |
> > | SIF + PCA5 + norm   | 0.6492 | 0.7183 | 0.7440 | 0.7631 |
> > +----------------------------+-----------+----------+----------+-----------+
> > Fair comparison of SIF and uSIF, using glove.840B.300d. The values are Pearson correlations. PCA is the principle component removal as per Arora et al. 2016 and PCA5 is the weighted principle component removal Ethayarajh, 2018.

---

### Meta-Review · Area_Chair1 · 2018-12-14
**Motivation and major contribution not clear.**

**Confidence:** 3
**Recommendation:** Reject

**Metareview:**

This paper presents a novel family of probabilistic approaches to computing the similarities between two sentences using bag-of-embeddings representations, and presents evaluations on a standard benchmark to demonstrate the effectiveness of the approach. While there seem to be no substantial disputes about the soundness of the paper in its current form, the reviewers were not convinced by the broad motivation for the approach, and did not find the empirical results compelling enough to serve as a motivation on its own. Given that, no reviewer was willing to argue that this paper makes an important enough contribution to be accepted.

It is unfortunate that one of the assigned reviewers—by their own admission—was not well qualified to review it and that a second reviewer did not submit a review at all, necessitating a late fill-in review (thank you, anonymous emergency reviewer!). However, the paper was considered seriously: I can attest that both of the two higher-confidence reviewers are well qualified to review work on problems and methods like these.